# Circulating Omentin-1, Sustained Inflammation and Hyperphosphatemia at the Interface of Subclinical Atherosclerosis in Chronic Kidney Disease Patients on Chronic Renal Replacement Therapy

**DOI:** 10.3390/medicina58070890

**Published:** 2022-07-02

**Authors:** Davide Bolignano, Marta Greco, Valentina Arcidiacono, Pierangela Presta, Alfredo Caglioti, Michele Andreucci, Francesco Dragone, Daniela Patrizia Foti, Giuseppe Coppolino

**Affiliations:** 1Renal Unit, “Magna Graecia” University, 88100 Catanzaro, Italy; arcidiaconovalentina@gmail.com (V.A.); piera.presta@gmail.com (P.P.); alfredocaglioti@libero.it (A.C.); andreucci@unicz.it (M.A.); gcoppolino@unicz.it (G.C.); 2Clinical Pathology Lab, “Magna Graecia” University, 88100 Catanzaro, Italy; marta.greco@unicz.it (M.G.); francesco.dragone@studenti.unicz.it (F.D.); foti@unicz.it (D.P.F.)

**Keywords:** Omentin-1, end-stage kidney disease, atherosclerosis, carotid intima–media thickness

## Abstract

*Background and Objectives*: Subclinical atherosclerosis, reflected by abnormal carotid intima–media thickness (cIMT), is pervasive among chronic kidney disease patients on chronic renal replacement therapy (RRT), being mostly influenced by uremia-related rather than traditional risk factors. *Materials and Methods*: In this pilot study, we measured circulating levels of Omentin-1, a recently discovered adipokine with strong anti-atherogenic properties, in a heterogeneous cohort of 77 asymptomatic RRT individuals (40 chronic kidney transplant recipients, Ktx; and 37 chronic hemodialysis patients, HD) and in 30 age-matched controls. *Results*: Omentin-1 was increased in RRT individuals as compared with controls (*p* = 0.03). When stratifying for renal replacement modality, we found Ktx patients to have significantly lower Omentin-1 than HD patients (*p* = 0.01). Lower Omentin-1 levels were also found among RRT individuals with pathological cIMT (168.7 [51.1–457.8] vs. 474.9 [197.2–1432.1]; *p* = 0.004). Our multivariate correlations analysis revealed Omentin-1 as the most robust independent predictor of carotid atherosclerosis (β-0.687; *p* = 0.03), even more than total cholesterol, diastolic BP and age, and this adipokine was at the crossroad of a complex interplay with sustained inflammation (high CRP and ferritin) and hyperphosphatemia in predicting higher cIMT values. *Conclusion*: The findings reported extend to renal patients with advanced disease, with the possible involvement of Omentin-1 in the pathogenesis of atherosclerosis. This may set the stage for future interventional studies of Omentin-1 replacement to retard atherosclerosis progression, as it is currently being investigated in other disease settings.

## 1. Introduction

Subclinical atherosclerosis, generally associated with abnormal carotid intima–media thickness (cIMT) values, is pervasive among chronic kidney disease patients on chronic renal replacement therapy (RRT) due to advanced renal disease, and impacts their cardiovascular (CV) risk profile [1]. In contrast to the general population, atherosclerosis may progress faster in RRT patients, as it is driven more by specific “uremia-related” factors, such as endothelial dysfunction, sustained inflammation and altered mineral metabolism [2].

Omentin-1 is a newly identified adipokine which is involved in various dysmetabolic conditions, including diabetes, metabolic syndrome and obesity, as well as in bone–mineral disorders and systemic inflammatory diseases [3]. Importantly, this adipokine exerts important antioxidative and anti-apoptotic activities and, above all, may suppress atherosclerosis progression by promoting normal vascular remodeling [4]. Altered Omentin-1 levels have previously been reported among end-stage kidney disease (ESKD) patients on chronic hemodialysis (HD) treatment [5], in whom it holds prognostic value for vascular disease progression [6] and CV mortality [7]. The relationship between this adipokine and subclinical atherosclerosis in the ESKD setting, however, has not yet been elucidated.

Starting from these premises, we thus designed a pilot observational study to cross-analyze circulating Omentin-1, clinical risk factors and the severity of carotid atherosclerosis in a cohort of asymptomatic individuals on chronic RRT.

## 2. Materials and Methods

From a source population of 99 adult uremic patients periodically referred to the Nephrology and Dialysis unit of the University Hospital of Catanzaro, Italy, we recruited 77 individuals on chronic renal replacement therapy. Of these, 37 were patients on stable HD treatment (RRT-HD) by high-flux bicarbonate dialysis, following a standard of 3 times/week regimen, while 40 were chronic kidney transplant recipients (RRT-Ktx) who were followed in an outpatient setting. We excluded individuals with unstable clinical conditions, malignancy, liver, thyroid or infectious diseases, recent acute CV episodes, infections, cancer, recent transplantation or dialysis onset (<6 months). The study was approved by the Local University Institutional Review Board, and all participating subjects provided written informed consent. Omentin-1 was measured in the blood by a commercially available ELISA kit (Human Intelectin-1/Omentin ELISA Kit, Novus Biologicals, Centennial, CO, USA), together with standard biochemical parameters. Blood pressure was calculated as the average of three consecutive measurements. Clinical, demographic and anthropometric data were recorded by using a standardized electronic case report form. Carotid intima–media thickness (cIMT) was measured in the posterior wall of both carotid arteries by mode B ultrasound and computed by measuring the thickness of the innermost two layers of intima–media, 5 mm before the bifurcation of the common carotid artery. Pathological (high) cIMT was assumed for mean (right/left) values > 0.9 mm and/or a unilateral cIMT over the 75th percentile of the established age- and sex-dependent reference ranges [8]. All clinical and laboratory measurements were made before starting a mid-week dialysis session in HD patients or upon arrival at the outpatient clinic for a scheduled visit in Ktx. Thirty age-matched healthy individuals served as controls for Omentin-1 testing.

The statistical analysis was performed by using the SPSS package (version 25.0; IBM corporation, Armonk, NY, USA) and the GraphPad prism package (version 8.4.2, GraphPad Software, San Diego, CA, USA). Differences between groups were assessed by the unpaired *t*-test for normally distributed values, the Mann–Whitney U test for non-parametric values and the chi-square, followed by a Fisher’s exact test for frequency distributions. The Pearson (R) and the Spearman (Rho) correlation coefficients were employed to identify the univariate correlates of cIMT. Skewed variables were preliminarily log-transformed to approximate normal distributions. Various multiple regression models considering cIMT as the dependent variable were built, including univariate correlates in different combinations. All results were considered significant for *p*-values ≤ 0.05.

## 3. Results

The main characteristics of the study population are summarized in Table 1. The mean age was 60 ± 14.1 years, and 67.5% of individuals were male. The prevalence of diabetes was 27.7%, and most subjects (70.1%) were hypertensive under treatment. The frequency of previous CV diseases ranged from 22 to 44.1%. In Ktx, the mean eGFR (CKD-Epi formula) was 64.8 ± 12.3 mL/min/m^2^. Omentin-1 levels were higher in the whole RRT cohort as compared with healthy controls (324 [90.3–770] vs. 110 [35.4–240.9] pg/mL; *p* = 0.03, Figure 1A); however, after stratification for renal replacement modality, no differences in such levels were noticed between RRT-Ktx and controls, while, on the contrary, higher Omentin-1 levels were found in RRT-HD patients (474.9 [197.2–1432.1] ng/mL) with respect to both healthy subjects (*p* = 0.009) and RRT-Ktx (*p* = 0.01) (Figure 1B).

The mean cIMT in the whole study cohort was 0.78 ± 0.32 mm. On average, cIMT values were abnormal in 36 patients (46.7%). Such individuals were older (*p* = 0.001) and displayed higher phosphate (*p* = 0.03), cholesterol (*p* = 0.001), hs-CRP (*p* = 0.001) and ferritin (*p* = 0.03) levels, as well as lower diastolic BP (*p* = 0.001) and higher pulse pressure (*p* = 0.009). Marginal, although not statistically significant, differences between the two subpopulations were noticed with respect to the prevalence of diabetes, glycemia, potassium, albumin, LDL cholesterol and platelet levels (*p*-value range 0.06–0.10). Omentin-1 was significantly lower in RRT individuals with pathological cIMT as compared to others (168.7 [51.1–457.8] vs. 474.9 [197.2–1432.1]; *p* = 0.004. Figure 2). Table 1 resumes differences between the two study subgroups. The univariate correlation analyses (Table 2) confirmed age (R = 0.329; *p* = 0.007), serum phosphate (R = 0.225; *p* = 0.04), total cholesterol (R = 0.398; *p* = 0.01), hs-CRP (R = 0.297; *p* = 0.01) and ferritin (R = 0.260; *p* = 0.03) as directly associated with cIMT values, while inverse correlations were found with diastolic BP (R = −0.250; *p* = 0.04) and Omentin-1 (R = −0.333; *p* = 0.006). Of note, during the exploratory subgroup analyses, no correlations were found between eGFR and Omentin-1 levels in Ktx.

In a fully adjusted multivariable model (Model 1) including all the univariate significant predictors of cIMT, the associations between the severity of carotid atherosclerosis and, respectively, Omentin-1, total cholesterol, diastolic BP and age remained significant, while those with ferritin, serum phosphate and hs-CRP were lost. Of note, in this model, Omentin-1 ranked as the strongest independent predictor of cIMT (β-0.687; *p* = 0.03). Significant associations between cIMT and inflammatory indexes (hs-CRP, ferritin) re-emerged after excluding Omentin-1 from the model (Model 2), while phosphate remained an independent predictor of cIMT only in a model simultaneously excluding Omentin-1 and inflammation (Model 4). Data from multivariate models are detailed in Table 2.

## 4. Discussion

The findings from this study raises a couple of prompts for discussion. First, although we found, on average, markedly increased Omentin-1 levels in all patients on chronic RRT as compared with healthy controls, lower Omentin-1 values were observed in a sub-cohort of RRT-Ktx with respect to patients on maintenance HD. In this regard, a modulatory effect exerted by the chronic immunosuppressant therapy could partially be called into question, given the well-known relationships between this adipokine and the overall activation of the immune system [3]. In addition, a partial recovery in renal clearance by kidney transplantation could also play a key role; however, although such a hypothesis would pair well with findings in chronic kidney disease patients with partially conserved renal function [9,10], no significant correlations were found at exploratory analyses between residual renal function (eGFR) and Omentin-1 in the subgroup of Ktx. Ultimately, the question as to whether lower Omentin-1 levels in high-risk populations could be considered detrimental rather than beneficial remains timely. Indeed, reduced Omentin-1 levels predict worse outcomes in diabetic patients [11], in individuals with recent ischemic stroke [12] and in those with severe cardiomyopathy [13]. No less important, although uremia may upregulate the circulating balance of Omentin-1, lower values predict worse outcomes also in this disease setting [6,7]. A possible explanation for such an apparent discrepancy might depend on a documented inverse relationship between this adipokine and the severity of atherosclerosis. In fact, in individuals with metabolic syndrome, diabetes or history of CV disease, circulating Omentin-1 levels are markedly downregulated in the presence of either subclinical or overt atherosclerotic disease, reflecting the severity at the carotid and coronary level [14,15,16,17]. Whether such reduced levels represent a causal factor, a compensatory response or just an epiphenomenon of the atherosclerotic process remains a much-debated issue.

Findings from our study confirm a close inverse association between Omentin-1 and the severity of subclinical carotid atherosclerosis, as assessed by cIMT measurement.

Although cIMT is a surrogate of incipient cardiovascular disease, its prognostic importance in the general population and in kidney disease patients as a robust risk stratifier for CV mortality remains well established [18]. Although Ktx often elicits a significant improvement in IMT as compared to patients remaining on maintenance dialysis, abnormally increased IMT persists in a large part of asymptomatic Ktx and may worsen over time to frank vascular disease [19].

A possible relationship between Omentin-1 and subclinical atherosclerosis in this miscellaneous cohort of chronic RRT patients relies on diverse observations. First, RRT patients with abnormal cIMT values, which represented almost one-half of the total study population, displayed markedly lower Omentin-1 values as compared with others. In this regard, a potential confounding effect by Ktx could be, in principle, ruled out, as such renal replacement modality displayed a similar frequency among patients with normal or pathological cIMT. By the same token, no associations were found during the correlation analysis between cIMT and HD or Ktx, while, on the contrary, a strong inverse relationship was found with circulating Omentin-1 levels. Data from multivariable analyses confirmed low Omentin-1 as an independent predictor of high cIMT in all the various models tested. Likewise, only the correlations with other robust risk factors—namely age, cholesterol and diastolic BP—remained unaffected by other confounders. Conversely, in targeted models of increasing complexity, inflammatory indexes (hs-CRP and ferritin) and hyperphosphatemia remained significantly associated with cIMT severity only if Omentin-1 was not introduced in the model.

Persistent inflammation is a renowned risk factor for progressive atherosclerosis in ESKD, as it elicits endothelial dysfunction, abnormal wall remodeling and microvascular damage [20]. Similarly, bone mineral disorders, altered fibroblast growth factor 23 (FGF23), low vitamin-D levels and hyperphosphatemia all impinge upon microvascular dysfunction, thereby triggering or worsening subclinical atherosclerosis [21]. Previous studies posed Omentin-1 at the crossroad of bone–vascular disease in systemic inflammatory conditions, in which a beneficial effect of this adipokine upon bone loss and vascular damage has clearly been demonstrated [22].

## 5. Conclusions

In light of these observations, Omentin-1 might reasonably be placed in the casual pathways linking, respectively, inflammation and hyperphosphatemia to the severity of carotid atherosclerosis in patients with chronic kidney disease needing chronic renal replacement therapy. In this view, this adipokine would represent one missing link of the pro-atherogenic effects of altered mineral balance and chronic inflammation, thereby representing a potential therapeutic target for replacement therapy, as it is currently being investigated in other disease settings [4].

## Figures and Tables

**Figure 1 medicina-58-00890-f001:**
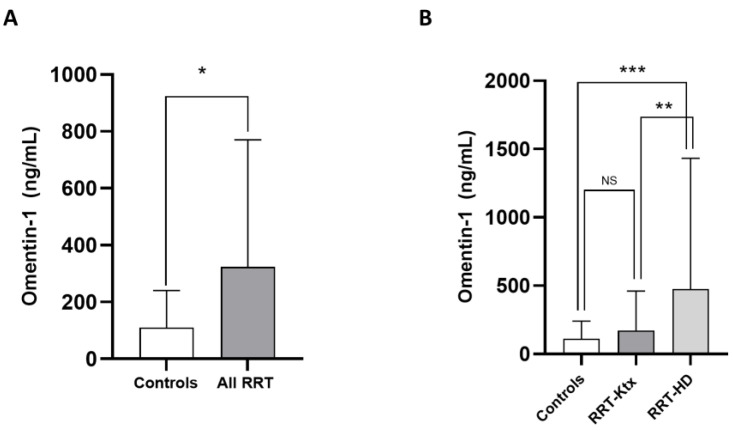
Differences in median circulating Omentin-1 between (**A**) healthy controls and all RRT patients and between (**B**) healthy controls, RRT-KTx patients and RRT-HD patients; * *p* = 0.03, ** *p* = 0.01 and *** *p* = 0.009.

**Figure 2 medicina-58-00890-f002:**
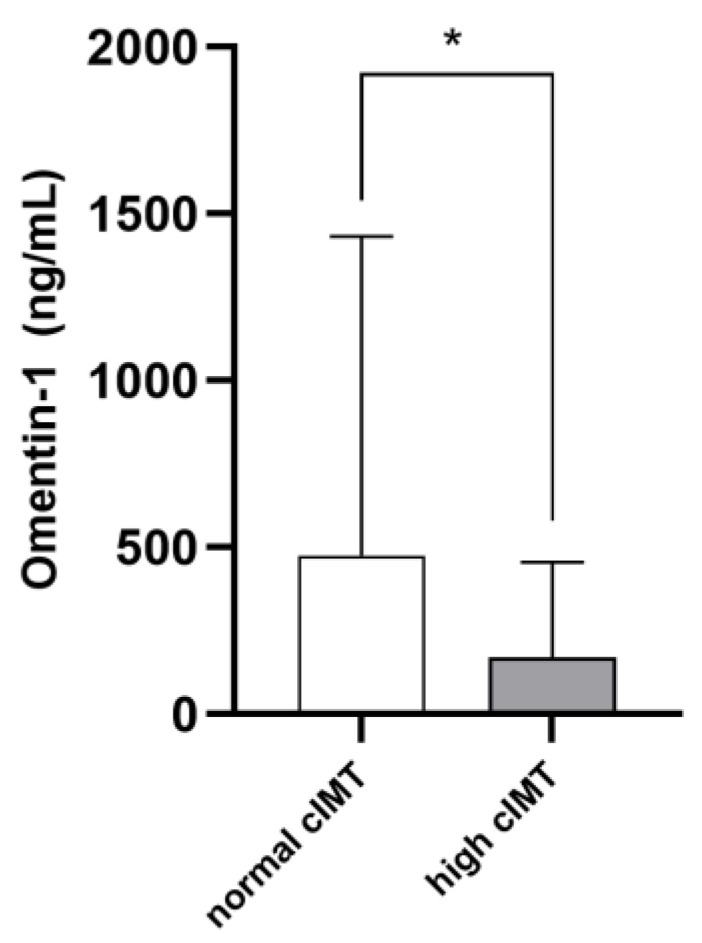
Median Omentin-1 levels in RRT patients with normal or high cIMT. * *p* = 0.004.

**Table 1 medicina-58-00890-t001:** Main clinical and laboratory characteristics in the whole study cohort and in subgroups of patients with normal or pathological cIMT. Statistical differences are highlighted in bold.

	All RRT Patients *N* = 77	Normal cIMT *N* = 41	High cIMT *N* = 36	*p*
Age (yrs)	60 ± 14.1	**57.2 ± 15.1**	**69.3 ± 14.1**	**0.001**
Males *n* (%)	52 (67.5)	25 (61)	27 (75)	0.31
HD/Ktx (*n*)	37/40	19/22	18/18	0.59
BMI (kg/m^2^)	26.0 ± 6.4	26.3 ± 5.8	25.9 ± 5.3	0.66
Waist-Hip ratio (cm)	0.93 ± 0.07	0.95 ± 0.09	0.92 ± 0.16	0.19
AMI/Angina *n* (%)	17 (22)	10 (24.4)	7 (19.4)	0.54
Hypertension *n* (%)	54 (70.1)	30 (73.1)	24 (66.7)	0.67
Heart failure *n* (%)	34 (44.1)	18 (43.9)	16 (44.4)	0.87
Diabetes *n* (%)	21 (27.7)	8 (19.5)	13 (36.1)	0.10
Glycemia (mg/dL)	102.7 ± 26.7	99.5 ± 13.2	114.3 ± 12.1	0.06
Hemoglobin (g/dL)	11.7 ± 2	11.8 ± 1.8	11.7 ± 1.2	0.72
SBP (mmHg)	138.7 ± 21.2	135.6 ± 14.6	140.1 ± 14.3	0.21
DBP (mmHg)	79.5 ± 12.7	**85.5 ± 11.3**	**72.0 ± 10.8**	**0.001**
Pulse pressure (mmHg)	60.4 ± 18.6	**51.3 ± 11.1**	**67.5 ± 13.6**	**0.009**
Serum creatinine (mg/dL)	2.9 [1.3–7.2]	3.5[1.9–5.8]	1.9[1.1–8.8]	0.36
Urea (mg/dL)	94.8 ± 46.1	97.3 ± 30.1	95[58–110]	0.38
Sodium (mg/dL)	139.6 ± 6.7	138.9 ± 11.4	139.3 ± 15.6	0.70
Potassium (mg/dL)	4.69 ± 0.67	4.05 ± 0.95	4.95 ± 0.60	0.09
Phosphate (mg/dL)	4.74 ± 1.39	**4.13 ± 0.85**	**5.48 ± 1.10**	**0.03**
Calcium (mg/dL)	9.39 ± 0.79	9.48 ± 0.62	9.31 ± 0.52	0.49
Magnesium (mg/dL)	2.21 ± 0.58	2.25 ± 0.45	2.16 ± 0.39	0.20
iPTH (pg/mL)	179.3[105.5–358]	95[79.8–188.5]	304.3[63.5–408]	0.08
nt-proBNP (pg/mL)	817 [137.7–3320.7]	525 [107.4–1320.1]	310 [203.2–4500]	0.46
Uric acid (mg/dL)	5.75 ± 1.35	5.44 ± 1.19	5.98 ± 1.45	0.33
Albumin (g/dL)	4.22 ± 0.39	4.30 ± 0.19	3.99 ± 0.31	0.07
Total Cholesterol (mg/dL)	167 ± 41.5	**149.1 ± 29.9**	**216.3 ± 29.5**	**0.001**
HDL (mg/dL)	53.2 ± 14.3	55.2 ± 12.9	51.8 ± 11.1	0.12
LDL (mg/dL)	100.2 ± 35.5	99.6 ± 33.8	103.3 ± 29.5	0.07
Triglycerides (mg/dL)	123 [85–168]	121 [58–206]	126 [43–199]	0.49
ESR (mm/h)	18 [9–30]	12 [8–84]	38 [20–56]	0.19
hs-CRP (mg/L)	3.23 [0.21–6.55]	**2.9 [0.21–3.65]**	**13.1 [3.23–25.1]**	**0.001**
Fibrinogen (mg/dL)	366.9 ± 103.2	364.4 ± 101.1	355 [95.6–422.9]	0.56
Ferritin (mcg/L)	97 [30–208]	**89.2 [20.6–108.2]**	**120.3 [54.1–350.3]**	**0.03**
RBC (*n* ×10^3^)	4.26 ± 1.05	4.25 ± 0.99	4.28 ± 0.78	0.69
WBC (*n* ×10^3^)	6.90 ± 2.08	6.80 ± 2.75	6.99 ± 3.01	0.21
PLT (*n* × 10^3^)	225.2 ± 86.3	229 ± 99.1	222.4 ± 58.6	0.10
cIMT (mm)	0.78 ± 0.32	**0.54 ± 0.16**	**1.05 ± 0.21**	**<0.001**
Omentin-1 (ng/mL)	324 [90.3–770]	**474.9 [197.2–1432.1]**	**168.7 [51.1–457.8]**	**0.004**

Legend: RRT, renal replacement therapy; cIMT, carotid intima–media thickness; BMI, body mass index; HD, hemodialysis; KTx, kidney transplantation; AMI, acute myocardial infarction; SBP, systolic blood pressure; DBP, diastolic blood pressure; iPTH, intact parathyroid hormone; nt-proBNP, n terminal pro-brain natriuretic peptide; HDL, high-density lipoprotein; LDL, low-density lipoprotein; ESR, erythrocyte sedimentation rate; hs-CRP, high-sensitivity c-reactive protein; RBC, red blood cell; WBC, white blood cell; PLT, platelets.

**Table 2 medicina-58-00890-t002:** Univariate correlates of cIMT in RRT patients and multivariate models employing cIMT as the dependent variable. Statistically significant associations are highlighted in bold.

*Univariate Associations*		
	**R**	** *p* **
**Total cholesterol**	**0.398**	**0.01**
**(log) Omentin-1**	**−0.333**	**0.006**
**Age**	**0.329**	**0.007**
**(log) hs-CRP**	**0.297**	**0.01**
**(log) Ferritin**	**0.260**	**0.03**
**Diastolic BP**	**−0.250**	**0.04**
**Phosphate**	**0.225**	**0.04**
***Multivariate Model 1—fully adjusted*** Multiple R = 0.71, R^2^ = 51%; *p* < 0.0001		
	** *β* **	** *p* **
**(log) Omentin-1**	**−0.687**	**0.03**
**Total cholesterol**	**0.415**	**0.001**
**Diastolic BP**	**−0.311**	**0.02**
**Age**	**0.247**	**0.02**
*(log) Ferritin*	*0.443*	*0.27*
*Phosphate*	*−0.051*	*0.61*
*(log) hs-CRP*	*0.255*	*0.10*
***Multivariate Model 2—excluding Omentin-1*** Multiple R = 0.69, R^2^ = 48%; *p* < 0.0001		
	** *β* **	** *p* **
**Total cholesterol**	**0.429**	**0.001**
**Age**	**0.280**	**0.01**
**(log) hs-CRP**	**0.272**	**0.007**
**Diastolic BP**	**−0.271**	**0.02**
**(log) Ferritin**	**0.226**	**0.04**
**Phosphate**	**0.190**	**0.04**
***Multivariate Model 3—excluding hs-CRP and Ferritin*** Multiple R = 0.65, R^2^ = 42%; *p* < 0.0001		
	** *β* **	** *p* **
**Total cholesterol**	**0.369**	**0.003**
**Diastolic BP**	**−0.301**	**0.01**
**Age**	**0.296**	**0.009**
**(log) Omentin-1**	**−0.289**	**0.02**
*Phosphate*	*0.099*	*0.54*
***Multivariate Model 4—excluding hs-CRP, Ferritin and Omentin-1*** Multiple R = 0.61, R^2^ = 37%; *p* < 0.0001		
	** *β* **	** *p* **
**Total cholesterol**	**0.522**	**<0.001**
**Age**	**0.280**	**0.01**
**Diastolic BP**	**−0.265**	**0.03**
**Phosphate**	**0.201**	**0.04**

Legend: BP, blood pressure; hs-CRP, high-sensitivity c-reactive protein.

## Data Availability

Raw data from this study can be shared by the corresponding author upon reasonable request.

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
