# Peer review of "Circulating Omentin-1, Sustained Inflammation and Hyperphosphatemia at the Interface of Subclinical Atherosclerosis in Chronic Kidney Disease Patients on Chronic Renal Replacement Therapy"

_medicina, 2022, doi:10.3390/medicina58070890_

Round 1

Reviewer 1 Report

My impression and suggestions are reported in the attached text file

Author Response

This paper outlines the possible role of omentin-1 in the subclinical atherosclerosis of chronic kidney disease (CKD) patients, as evaluated by carotid intima-media thickness (cIMT) and explores its relationship with other known risk factors and/or parameters in this special population, like hyperphosphatemia, inflammation, total cholesterol, diastolic blood pressure and age. It is a cross sectional study, therefore etiologic implications cannot be drawn, however the paper attempts to enlighten the involvement of Omentin-1 in this pathology.

General comment

The language is clear and easy to read, and the message is current. It is a cross sectional and observational study, so the conclusions are limited. In terms of methodology, some aspects must, in my opinion, be improved and they will be named in the following lines.

Specific comments

Abstract: describes the general study and is globally well written. Nonetheless, it can be improved. Material and methods: the authors refer that they evaluated a group of end-stage kidney disease (ESKD) patients in whom serum Omentin-1 levels were analysed. Within the “results” of the abstract - and later in the article - the reader realizes that the same evaluation was also performed in a control group of healthy volunteers. So, this relevant information should be put in the Material and methods also in the abstract.

R- We thank the reviewer for this suggestion. This information has now been added in the abstract as well.

As a suggestion I would not call kidney transplant recipients “ESKD” patients, for their renal function is not comparable with the CKD stage 5D patients on hemodialysis.

R- we thank the reviewer for this important suggestion. In order to avoid misunderstanding we have now replaced the term ESKD with RRT (renal replacement therapy – as either hemodialysis and Ktx could be considered as RRT) in the whole manuscript. The title has also been changed accordingly.

Introduction: It is clear and concise and gives an overview of what is aimed to be evaluated.

Material and methods: some data should be clarified.

It is said that “37 were patients on stable HD treatment (ESKD-HD) by a standard 3 times/week”. The modality of hemodialysis should be specified – was this high-flux or low-flux dialysis or on-line hemodiafiltration? This may be relevant, for Omentin-1 is a 34KD molecule, and is hardly removed by conventional hemodialysis, but the inference of the results must be done in this specific population, that was performing hemodialysis in a specific manner (1,2).

R- all patients recruited in this study were on conventional hemodialysis. This information has now been specified. We acknowledge the possible impact of alternative HD techniques on Omentin-1 removal and, indeed, results from this pilot study are setting the stage for future, planned investigations, among which the idea of comparing the impact of HD, HDF and other techniques (HFR, AFB and, lastly, CRRTs) on Omentin-1 balance.

“Omentin-1 was measured (…) together with standard biochemical parameters”. In the hemodialysis population when was Omentin-1 and the other laboratory items measured? Pre dialysis? Before the midweek session? Was it the same way for all individuals under hemodialysis? This seems important for, in theory, the timing of laboratory evaluation (pre or post HD; number of the dialysis in the week) of a patient under dialysis may influence the laboratory blood results.

R- All measurements were made before starting a single, mid-week dialysis session in HD patients or before a scheduled visit in Ktx recipients. We have now provided this information.

Also, it is expressed that “Main clinical, demographic and anthropometric data were recorded using a standardized, electronic case report form”. Which blood pressure was considered in the ESKD-HD population? Was it considered the pre dialysis blood pressure? Was it considered the mean of several measurements? of which HD session? This data should be specified.

R- BP assessment was made before starting a mid-week dialysis session in HD patients or upon arrival at the outpatient clinic for a scheduled visit in Ktx recipients. BP was calculated as the average of three consecutive measurement. We have now added this information

The description of the statistical analysis seemed appropriate.

Results: several items should be elucidated.

It is said also that the entire group of ESKD patients had higher Omentin-1 levels than the control group (“324 [90.3-770] vs. 110 [35.4-240.9] pg/mL”). In figure 1a, the values of ESKD patients appear to be higher than 500 ng/mL. Therefore, there is a clear incongruence between the figure values and the ones mentioned in the text itself.

It is also referred that “higher Omentin-1 was found in ESKD-HD patients (474.9 [197.2-1432.1] ng/mL) with respect to healthy subjects (p=0.009) and ESKD-Ktx (p=0.01) (Figure 1b)”. In this figure, the values depicted in this population (ESKD-HD) are well over 500 ng/mL, the ones ESKD-tx approximately 400 ng/mL and the normal population around 250 ng/mL. So, the information in the text and in figures doesn’t coincide.

It is also claimed that “Omentin-1 was significantly lower in ESKD with pathological cIMT as compared to others (168.7 [51.1-457.8] vs. 474.9 [197.2-1432.1]; p=0.004. Figure 2).” Again, the values depicted in the figure are quite different from the values expressed in the text, with the population with normal cIMT showing values well over 700 ng/mL and pathological cIMT population with values around 250 ng/mL.

R-We acknowledge this apparent incongruence, and we thank the reviewer for raising this point. Actually , as indicated in the legend, figures showed Omentin-1 data as mean+/- S.E. while in the plain text, information and analyses were provided as median [IQR]. In order to avoid misunderstanding we have now reformatted the figures to show median instead of mean values of Omentin-1.

Within the text it is reported that 36 patients (46,7%) had abnormal cIMT but in table 1 there were 37 (19 KTx and 18 HD) patients with abnormal cIMT (48,05% of ESKD). Additionally, if 48.6% of the studied population in HD (18/37 patients) has abnormal cIMT and these show lower values in comparison with the others ESKD, if the ESKD-tx show values comparable with the healthy control group, then, only 19 or 52% of the HD patients, are responsible for the higher Omentin-1 values that was seen in ESKD patients in comparison with controls. This appears to be mathematically difficult to explain, and, if this is the case, the values must be clearly expressed.

R- There was actually a typo error which has now been corrected and we thank the reviewer for identifying this. Indeed, high cIMT was found in 36 patients, which were equally distributed between Ktx and HD (18/18). We acknowledge the point raised by the reviewer about numbers and possible meanings. Unfortunately, we feel that any hypothesis, inference or even final interpretation on the possible meaning/single behaviors of Omentin-1 in patients stratified either by RRT modality and cIMT are significantly hampered by the limited sample size and by the pilot nature of the study itself. For this reason, we would avoid further sub-stratifications or sub-comparisons between patients and between patients and controls as the analyses would come up being inconclusive due to the limited statistical power.

As Omentin-1 levels seem to be influenced by the degree of renal function or CKD stage, the level of renal function in the KTx population should be indicated, in order to adequately interpret the results (3).

R- We have now introduced this information in the plain text. We preferred avoiding inserting it also in table 1 as data summarizes information from the whole uremic population (thus including also 37 pts who were on HD treatment and, therefore, with no eGFR available). Importantly, at exploratory subgroup analyses, no correlations were found between eGFR and Omentin-1 levels in Ktx recipients. This has been mentioned in the results and in the discussion section as well.

Regarding the correlations, caution should be taken when interpreting results: the study is a picture in time, so those were the values at that moment. There is no information about therapeutics, namely with iron or with phosphate binders in this population, which might have influenced respectively ferritin or phosphorus levels. These therapeutics are relatively common in hemodialysis patients. Although systolic and diastolic blood pressure values were both analysed, we found no information about pulse pressure which is a known cardiovascular risk factor, even in ESKD (4).

R-We thank the reviewer for this comment. We preferred to do not record and analyze information on therapeutics as in a large percentage of HD patients some therapies were recently adjusted, implemented or stopped. Nevertheless, we do acknowledge the potential important impact of therapeutics on modulating Omentin-1 levels and, for this reason, we are carefully taking into account this aspect in a future extended study that is currently under development. Conversely, following the reviewer’s suggestion, data on pulse pressure have now been introduced in the table.

In my opinion, it is not necessary to show all the models that were used in the multivariable analysis. There appears to be too much information without any real benefit to the reader. In table 2, apart from the univariate analysis, I would only show model 1 of multivariate analysis with Omentin-1.

R- We thank the reviewer for raising this point that is, of course, more than reasonable. Nevertheless, we feel that depicting the various multivariable models could, on the contrary, help the readership to better understand why Omentin-1 could be placed, in this population, at the crossroad of inflammation, hyperphosphatemia and incipient atherosclerosis. In fact, data from multivariable analyses confirmed low Omentin-1 as an independent predictor of high cIMT in all the various models tested, also in those including important confounders. Conversely, in targeted models of increasing complexity, inflammatory indexes (hs-CRP and ferritin) and hyperphosphatemia remained significantly associated with cIMT severity only if Omentin-1 was not introduced in the model. This supports the opening interpretation and, to our opinion, is important to be extensively showed in the table.

Discussion: appears to be adequate to the mentioned results, but those must be clarified….

The authors state that they noticed, according to others, higher Omentin-1 levels in CKD patients than in the healthy controls and that those values were comparable to the healthy population in kidney transplant recipients. A partial recovery of renal function might explain those facts. Indeed, this was seen in other studies. However, to my knowledge, no study has shown the reason why Omentin-1 levels are higher in later stages of CKD and in dialysis patients. The reasoning seems appropriate and not too emphatic. But, as previously mentioned, the kidney function of KTx is not shown in this paper. In conclusion, the entire results of the study must be clarified.The whole discussion seems clear and logical, but can’t be applicable to the study results, as they were shown here.

R- As previously suggested, we have now provided information on residual kidney function in the subpopulation of Ktx recipients. We take again the opportunity to underline that, at exploratory analyses no correlation has been found between Omentin-1 and eGFR in this subcohort. This might be explained by a true lack of relationship or by the fact that the study was underpowered to answer this question. Nevertheless, we would like to recall again the pilot and observational nature or the investigation and the concise design of this paper (brief report) which is not aiming at providing conclusive interpretations or clarifying the exact biological meanings of observations but is rather finalized at generating hypothesis and setting the stage for future, targeted investigations that are currently under development.

Conclusion:

It is a generalized statement, and nothing else could be said concerning a cross sectional study.

References: seem adequate

Figures and tables: need improvement - the comments were already performed in the above sections

R- All comments have previously been addressed

Reviewer 2 Report

To the authors

Thank you for the opportunity to review this article.

In this study, the authors showed the cross-analysis of circulating Omentin-1, clinical risk factors, and carotid atherosclerosis severity. As a result, omentin-1 was the most robust independent predictor of carotid atherosclerosis. This report is very interesting. However, several problems should be resolved for the manuscript to be accepted for publication in Medicina.

<Major points>

1. In this study, both hemodialysis patients and kidney transplant patients are analyzed as ESRD. However, the cardiovascular risk of hemodialysis and kidney transplant patients is very different (PMID: 21883901). The authors should also provide data on the relationship between cIMT and Omentin-1 in HD patients and renal transplant patients, respectively.

2. This study compares Omentin-1 in ESRD with healthy subjects. However, Omentin-1 in CKD patients is also important. The authors should add the data about omentin-1 in CKD patients in the result section or a relevant reference in the discussion section.

<Minor points>

3. I am concerned that immunosuppressive therapy may affect Omentin-1 in kidney transplant patients. The authors need to discuss this topic.

4. cIMT is used as an outcome in this study. However, cIMT is a surrogate marker for cardiovascular events. More careful discussion is needed regarding the use of cIMT as an endpoint.

Author Response

Thank you for the opportunity to review this article.

In this study, the authors showed the cross-analysis of circulating Omentin-1, clinical risk factors, and carotid atherosclerosis severity. As a result, omentin-1 was the most robust independent predictor of carotid atherosclerosis. This report is very interesting. However, several problems should be resolved for the manuscript to be accepted for publication in Medicina.

<Major points>

  1. In this study, both hemodialysis patients and kidney transplant patients are analyzed as ESRD. However, the cardiovascular risk of hemodialysis and kidney transplant patients is very different (PMID: 21883901). The authors should also provide data on the relationship between cIMT and Omentin-1 in HD patients and renal transplant patients, respectively.

R- We thank the reviewer for raising this important point that we fully acknowledge and agree with. Unfortunately, the limited sample size of the two subpopulations prevented us to perform such analyses in a reliable way, as the study resulted in both cases underpowered. Nevertheless, we fully agree with the importance of differentiate the risk profile in these two settings and , for this reasons, starting from these preliminary findings, we are now developing a more targeted (and more powered) study to answer also this research question.

  1. This study compares Omentin-1 in ESRD with healthy subjects. However, Omentin-1 in CKD patients is also important. The authors should add the data about omentin-1 in CKD patients in the result section or a relevant reference in the discussion section.

R- Omentin-1 balance in CKD patients has previously been investigated by few studies, which provided, in general, contradictory findings (see for instance refs 9-10-15 mentioned in our paper). This is mostly related to the presence of various comorbidities such as diabetes, obesity or frank peripheral vasculopathy that may affect circulating Omentin-1 balance in a more relevant way than the reduced renal function itself. For this reason, a control group represented by CKD patients in our study would have been detrimental and misleading for the analyses as it probably needed to be larger than the investigation group itself. Nevertheless, we acknowledge the importance of studying this adipokine across the whole CKD spectrum but we think that it should be addressed by targeted studies in very homogeneous populations

<Minor points>

  1. I am concerned that immunosuppressive therapy may affect Omentin-1 in kidney transplant patients. The authors need to discuss this topic.

 R- We fully agree with this possibility that has now been acknowledged in the discussion section.

  1. cIMT is used as an outcome in this study. However, cIMT is a surrogate marker for cardiovascular events. More careful discussion is needed regarding the use of cIMT as an endpoint.

R- We fully agree with this observation. We have now implemented a quick discussion on this issue and added two references (18-19) pertaining the importance of cIMT measurement as a risk stratifier.

Round 2

Reviewer 1 Report

Comment on

CIRCULATING OMENTIN-1, SUSTAINED INFLAMMATION AND HYPERPHOSPHATEMIA AT THE INTERFACE OF SUBCLINICAL ATHEROSCLEROSIS IN PATIENTS ON CHRONIC RENAL REPLACEMENT THERAPY

General comments

In my opinion the adjustments that were made in the article improved its quality.

I have some suggestions: since the studied population has chronic kidney disease, I would include this entity in the title, instead of “patients on chronic renal replacement therapy”.

In my view, it would be better to subdivide this CKD population in kidney transplant recipients (KTR) or hemodialysis patients (HDp).

Regardless, the authors call the population with chronic kidney disease, “on chronic renal replacement therapy (RRT)” and subdivide them as kidney transplant recipients (Ktx) and chronic hemodialysis patients (HD). Throughout the text however, the term “Ktx” or “Ktx recipients” is used several times concerning the same entity. So, I suggest standardizing this nomenclature throughout the text.

Abstract

Line 11: WhereMultivariate correlations analyses revealed” is written, I would say Multivariate correlation analysis revealed.

Introduction

Line 4: instead ofUnlike the general population, however, atherosclerosis may progress faster in RRT patients”, I would say Unlike the general population, atherosclerosis may progress faster in RRT patients,

Line 13: Instead ofit holds prognostic values”, it should be it holds prognostic value

Material and methods

Line 3: Regarding the sentencepatients on stable HD treatment (RRT-HD) by bicarbonate dialysis, following a standard 3 times/week regimen”, I suggest patients on stable (low or high-flux) HD treatment (RRT-HD) following a standard 3 times/week regimen

NOTE: it may be important, as I previously mentioned, the type of hemodialysis treatment that is performed. Is it a low-flux or high-flux hemodialysis?

Line 17: within the sentence “cIMT was assumed 80 for mean (dx/sx) values>0.9 mm and/or a unilateral cIMT”, I believe the authors want to refer to right/left side in “(sx/dx)”…

Results

Paragraph 2, line 4: “and higher pulse pressure (p=0.09)”, I believe one can only say “and tended to show higher pulse pressure (p=0.09)”. I would recommend, however, mentioning such fact in the following sentence “Marginal although not statistically significant differences between the two subpopulations were noticed with respect to the prevalence of diabetes, pulse pressure, glycemia, potassium…”, since even the value of “p” is within the referred interval in that sentence “(p range 0.06-0.10)”

Discussion

Third paragraph, line1: “Although cIMT is surrogate” I would say (…) is a surrogate

Fourth paragraph, line7: “no associations were found at correlation analyses between cIMT and HD or Ktx while” should be at correlation analysis.

Conclusion

I would be cautious considering that the whole population of chronic kidney patients has “advanced disease”, since “the mean GFR (CKD-Epi formula) was 64.8±12.3 mL/min/m2” in the kidney transplant recipients, which denotes near normal kidney function, as evaluated by this method.

Final note

 In my opinion this article may be published with the minor proposed alterations, which, in my view, would improve it.

Author Response

In my opinion the adjustments that were made in the article improved its quality.

I have some suggestions: since the studied population has chronic kidney disease, I would include this entity in the title, instead of “patients on chronic renal replacement therapy”.

In my view, it would be better to subdivide this CKD population in kidney transplant recipients (KTR) or hemodialysis patients (HDp).

R-As suggested, we have included the term chronic kidney disease in the title. However, we feel as important to keep the definition of “patients on chronic replacement therapy” both in the title and all along the manuscript as both the two subpopulation categories (HD and Ktx) fulfil this definition.

Regardless, the authors call the population with chronic kidney disease, “on chronic renal replacement therapy (RRT)” and subdivide them as kidney transplant recipients (Ktx) and chronic hemodialysis patients (HD). Throughout the text however, the term “Ktx” or “Ktx recipients” is used several times concerning the same entity. So, I suggest standardizing this nomenclature throughout the text.

R-In order to avoid misunderstanding we have removed the term “recipients” from the manuscript.

Abstract

Line 11: Where “Multivariate correlations analyses revealed” is written, I would say Multivariate correlation analysis revealed.

R-We have corrected this

Introduction

Line 4: instead of “Unlike the general population, however, atherosclerosis may progress faster in RRT patients”, I would say Unlike the general population, atherosclerosis may progress faster in RRT patients,

R-“however” has been removed

Line 13: Instead of “it holds prognostic values”, it should be it holds prognostic value

R- this has now been corrected

Material and methods

Line 3: Regarding the sentence “patients on stable HD treatment (RRT-HD) by bicarbonate dialysis, following a standard 3 times/week regimen”, I suggest patients on stable (low or high-flux) HD treatment (RRT-HD) following a standard 3 times/week regimen

NOTE: it may be important, as I previously mentioned, the type of hemodialysis treatment that is performed. Is it a low-flux or high-flux hemodialysis?

R- we have now specified that the treatment was high-flux HD

Line 17: within the sentence “cIMT was assumed 80 for mean (dx/sx) values>0.9 mm and/or a unilateral cIMT”, I believe the authors want to refer to right/left side in “(sx/dx)”…

R- yes, that’s correct, we have modified this.

 Results

Paragraph 2, line 4: “and higher pulse pressure (p=0.09)”, I believe one can only say “and tended to show higher pulse pressure (p=0.09)”. I would recommend, however, mentioning such fact in the following sentence “Marginal although not statistically significant differences between the two subpopulations were noticed with respect to the prevalence of diabetes, pulse pressure, glycemia, potassium…”, since even the value of “p” is within the referred interval in that sentence “(p range 0.06-0.10)”

R- We thank the reviewer for noticing this. It was actually a typo error (it was meant to be “p=0.009” instead of “p=0.09”). Indeed, the difference in pulse pressure was highly significant. We have corrected this also in the summary table.

 Discussion

Third paragraph, line1: “Although cIMT is surrogate” I would say (…) is a surrogate

R- We have corrected this

Fourth paragraph, line7: “no associations were found at correlation analyses between cIMT and HD or Ktx while” should be at correlation analysis.

R-This has also been corrected

 Conclusion

I would be cautious considering that the whole population of chronic kidney patients has “advanced disease”, since “the mean GFR (CKD-Epi formula) was 64.8±12.3 mL/min/m2” in the kidney transplant recipients, which denotes near normal kidney function, as evaluated by this method.

R-In order to avoid misunderstanding, we have removed the term “advanced”.

 Final note

 In my opinion this article may be published with the minor proposed alterations, which, in my view, would improve it.

Reviewer 2 Report

Thank you for your revised manuscript and response. The almost comments are well addressed. However, the problem described below should be resolved

Regarding major point 1, I understand the author’s concerns about the sample size. However, even with a small sample size, the univariate analysis is possible for the two subpopulations. Analysis should be presented to resolve concerns about the differences between hemodialysis and kidney transplant patients.

Author Response

Regarding major point 1, I understand the author’s concerns about the sample size. However, even with a small sample size, the univariate analysis is possible for the two subpopulations. Analysis should be presented to resolve concerns about the differences between hemodialysis and kidney transplant patients.

R-We fully agree with this concern and we are aware that the size of the two subpopulation would not, in principle, rule out the possibility of running such analyses. However, as pointed out previously, we have performed exploratory correlation analyses which did not provide significant results in terms of statistical significance, although a slight tendency in correlations has been noticed. With all probabilities, this can be attributed to a limited sample size of the two subpopulations. An already planned larger study will also aim at exploring this aspect.